# Disease-Resistance Functional Analysis and Screening of Interacting Proteins of ZmCpn60-3, a Chaperonin 60 Protein from Maize

**DOI:** 10.3390/plants14131993

**Published:** 2025-06-30

**Authors:** Bo Su, Lixue Mao, Huiping Wu, Xinru Yu, Chongyu Bian, Shanshan Xie, Temoor Ahmed, Hubiao Jiang, Ting Ding

**Affiliations:** 1Key Laboratory of Biology and Sustainable Management of Plant Diseases and Pests of Anhui Higher Education Institutes, School of Plant Protection, Anhui Agricultural University, Hefei 230036, China; subowsm31415@163.com (B.S.); maolixue1997@163.com (L.M.); wuhuiping2025@163.com (H.W.); y1902359894@163.com (X.Y.); bianchongyu@163.com (C.B.); 12116088@zju.edu.cn (H.J.); 2Anhui Province Key Laboratory of Integrated Pest Management on Crops, Hefei 230036, China; 3School of Biological and Food Engineering, Su Zhou University, Suzhou 215006, China; 4National Engineering Laboratory of Crop Stress Resistance Breeding, School of Life Sciences, Anhui Agricultural University, Hefei 230036, China; xssflora871216@126.com; 5Department of Plant Biotechnology, Korea University, Seoul 02481, Republic of Korea; temoorahmed@zju.edu.cn; 6Department of Life Sciences, Western Caspian University, Baku AZ1001, Azerbaijan

**Keywords:** maize, ZmCpn60-3, chaperonin 60 protein, disease-resistant, salicylic acid, interaction protein

## Abstract

Chaperonin 60 proteins plays an important role in plant growth and development as well as the response to abiotic stress. As part of the protein homeostasis system, molecular chaperones have attracted increasing attention in recent years due to their involvement in the folding and assembly of key proteins in photosynthesis. However, little is known about the function of maize chaperonin 60 protein. In the study, a gene encoding the chaperonin 60 proteins was cloned from the maize inbred line B73, and named *ZmCpn60-3*. The gene was 1, 818 bp in length and encoded a protein consisting of 605 amino acids. Phylogenetic analysis showed that ZmCpn60-3 had high similarity with OsCPN60-1, belonging to the β subunits of the chloroplast chaperonin 60 protein family, and it was predicted to be localized in chloroplasts. The *ZmCpn60-3* was highly expressed in the stems and tassels of maize, and could be induced by exogenous plant hormones, mycotoxins, and pathogens; Overexpression of *ZmCpn60-3* in *Arabidopsis* improved the resistance to *Pst* DC3000 by inducing the hypersensitive response and the expression of SA signaling-related genes, and the H_2_O_2_ and the SA contents of *ZmCpn60-3*-overexpressing *Arabidopsis* infected with *Pst* DC3000 accumulated significantly when compared to the wild-type controls. Experimental data demonstrate that flg22 treatment significantly upregulated transcriptional levels of the PR1 defense gene in *ZmCpn60-3*-transfected maize protoplasts. Notably, the enhanced resistance phenotype against Pseudomonas syringae pv. tomato DC3000 (*Pst* DC3000) in *ZmCpn60-3*-overexpressing transgenic lines was specifically abolished by pretreatment with ABT, a salicylic acid (SA) biosynthetic inhibitor. Our integrated findings reveal that this chaperonin protein orchestrates plant immune responses through a dual mechanism: triggering a reactive oxygen species (ROS) burst while simultaneously activating SA-mediated signaling cascades, thereby synergistically enhancing host disease resistance. Additionally, yeast two-hybrid assay preliminary data indicated that ZmCpn60-3 might bind to ZmbHLH118 and ZmBURP7, indicating ZmCpn60-3 might be involved in plant abiotic responses. The results provided a reference for comprehensively understanding the resistance mechanism of *ZmCpn60-3* in plant responses to abiotic or biotic stress.

## 1. Introduction

Proteins may experience misfolding and aggregation during the synthesis and folding process. In order to avoid misfolding of proteins, cells have evolved molecular chaperone networks. As a part of the protein homeostasis system, the molecular chaperone has attracted more attention due to its ability to participate in the folding and assembly of key proteins in photosynthesis in recent years. Researches have shown that molecular chaperones are ubiquitous in eukaryotes and prokaryotes, and there are two main types: Type I and Type II [1,2,3]. Type I existed in prokaryotes or organelles that originated from prokaryotes, including GroEL in *Escherichia coli*, Hsp60 in mitochondria and Cpn60 in chloroplasts [3,4], while Type II existed in archaea and eukaryotes, including thermosomes and CCT/TRiC [5,6].

Chaperonin 60 protein (Cpn60), as a type of molecular chaperone, is commonly known due to its size of approximately 60 KDa and mainly localized in chloroplasts. Studies have shown that Cpn60 could assist newly synthesized or immature peptides to fold and assemble by utilizing ATP hydrolysis [7,8], and is composed of two different types of subunits, Cpn60α and Cpn60β, which share approximately 50% homology between the two protein subunits [9,10,11,12]. Much research has indicated that Cpn60 protein directly affects protein folding and functional recovery, thereby regulating plant development and metabolic processes. It was found that a *Cpn60 α1* mutation caused *Arabidopsis* embryonic developmental defects and reduced the height of plant cotyledons [13]; *Cpn60 β1* mutation could lead to cell death in *Arabidopsis* leaves [14]; and the double mutation of *Cpn60 β1* and *Cpn60 β2* could lead to the lethal phenotype [15]. Moreover, a variety of substrates, including Rubisco, NDH complexes, and Rubisco activase, which play a crucial role in photosynthesis, need be folded and assembled with Cpn60 assistance [16]; Salvucci et al. also found that Rubisco activase could associate with Cpn60β to protect it from thermal denaturation during heat stress [17].

In recent years, with the depth of research, more and more reports have found that chaperonins (including Cpn60, HSP60, etc.) also play an important role in the process of plant stress resistance [13,18,19,20]. For example, the At*Cpn60β1* deletion could lead *Arabidopsis* seedling death and sensitivity to heat stress under short-day conditions [14]. Research demonstrates that *AtHSP17.6A* expression is specifically induced under heat shock, osmotic stress, and during seed developmental stages. Furthermore, transgenic Arabidopsis lines overexpressing this chaperone protein exhibit significantly enhanced adaptive capacity to combined salinity and drought stress conditions [21]. Moreover, overexpression of *OsCpn60β1* in *Oryza sativa* enhanced the photosynthesis and its resistance to salt stress. However, *OsCpn60β*1 loss-of-function mutants exhibited chloroplast developmental defects, manifesting an albino foliage and complete seedling lethality during early post-germinative growth stages [22]. At present, it has been shown that Cpn60 consists of two kinds of α subunits and four kinds of β subunits in the *Arabidopsis* chloroplast. Meanwhile, there are three types of α subunits and three types of β subunits in *Oryza sativa*. Researchers have conducted a series of studies around the above genes [14,22]. However, it is still unclear whether Cpn60 protein in maize can also play an important role in plant resistance to biotic or abiotic stress. Now, only a few small heat-shock protein genes such as *HSP16.9*, *HSP17.0*, *HSP17.3*, *HSP18.2*, and *HSP22* have been reported in maize. Overexpression of these heat-shock protein genes could enhance the heat resistance or hormone sensitivity of transgenic plants [23,24,25,26].Therefore, cloning and studying the function of maize *Cpn60* is of great significance for further understanding the role of chaperonin protein 60 in maize response to biotic or abiotic stress.

Previous studies demonstrated that the antagonistic endophyte *Bacillus subtilis* DZSY21 enhances maize resistance against *Bipolaris maydis*. Analysis of whole-genome bisulfite sequencing identified several resistance-related genes, including the candidate gene *Zm00001d054089_T001*, designated as *ZmCpn60-3* [27,28]. In this study, we isolated *ZmCpn60-3* from maize, analyzed the physicochemical properties and phylogenetic status of ZmCpn60-3 by the bioinformatics method, clarified the tissue expression and induction patterns of *ZmCpn60-3*, and elucidated its resistance mechanism in response to *Pst* DC3000 infection. Moreover, yeast two-hybrid and BIFC techniques were used to reveal the ZmCpn60-3 protein regulation and interaction network. Our findings were of great significance for comprehensively understanding the resistance mechanism of *ZmCpn60-3* and cultivating disease-resistant plant varieties.

## 2. Results

### 2.1. Identification and Cloning of ZmCpn60-3

The CDS of *ZmCpn60-3* (Zm00001d054089_T001) was obtained through genome release information (https://plants.ensembl.org/Zea_mays/Info/Index, accessed on 20 October 2019) and the coding region of *ZmCpn60-3* was amplified by RT-PCR using a cDNA template derived from maize RNA (Appendix A). Sequencing analysis revealed that ZmCpn60-3, localized to chromosome 4 in maize (*Zea mays*), comprises 1818 base pairs encoding a 605-amino-acid polypeptide. The encoded chaperonin protein exhibits a predicted isoelectric point (pI) of 5.81 and molecular mass of 64.36 kDa, containing a canonical Cpn60 conserved domain characteristic of GroEL-like chaperonin complexes (Appendix A).

Then, the phylogenetic tree of *ZmCpn60-3* was constructed using the genes of maize, *Oryza sativa*, and *Arabidopsis*, and shown in Figure 1. Through phylogenetic comparison, it was found that the *ZmCpn60-3* displayed similarities with maize *ZmCPN60-1* (NP001354057.1), *ZmCPN60-2* (PWZ25640.1), *ZmCPN60-40* (ACG36615.1), *ZmCPN60-5* (AQK61254.1), and *ZmCPN60-4* (ACG24216.1), and it had a close relationship with *OsCPN60-1* (XP 015626534.1) from *Oryza sativa* (Figure 1). The NCBI website predicts that the proteins, including *ZmCPN60-1*, *ZmCPN60-2*, *ZmCPN60-40*, *ZmCPN60-5*, *ZmCPN60-4*, and *OsCPN60-1*, all belonged to the β subunits of the chaperonin protein 60 family and were predicted to be located in chloroplasts; therefore, we speculated that *ZmCpn60-3* (*Zm00001d054089_T001*) is grouped into the β subunit. Subcellular localization prediction using WoLF PSORT (https://wolfpsort.hgc.jp/, accessed on 20 October 2023) indicated that ZmCpn60-3 localizes to the chloroplast.

### 2.2. Tissure Expression and Induction Patterns of ZmCpn60-3

The plant response to biotic stress is mainly regulated by hormone signals [29]. Therefore, the induced expression of *ZmCpn60-3* in maize treated with exogenous hormones was analyzed, and it was found that the expression levels of *ZmCpn60-3* in SA-treated and JA-treated maize both had a tendency of first increasing and then decreasing (Figure 2a,b); *ZmCpn60-3* expression reached its maximum level at 3 h post-SA treatment and 6 h post-JA treatment, which was about 2.45 and 7.80 times higher than that at 0 h, respectively (Figure 2a,b). However, the expression of *ZmCpn60-3* reached its peak at 12 h after ET application, which was approximately 1.84 times higher than that at 0 h, then decreased (Figure 2c). The above results showed *ZmCpn60-3* had an early response to SA (3 h), followed by JA (6 h), and a late response to ET (12 h), suggesting *ZmCpn60-3* may have different response patterns to these hormones.

Subsequently, the transcription level of *ZmCpn60-3* in maize treated with pathogens or toxins was analyzed. The expression of *ZmCpn60-3* in *Curvularia lunata*-treated plants at 12 h and *Pantoea stewartii*-treated plants at 24 h were 2.50-fold (Figure 2d) and 1.40-fold (Figure 2e) higher compared to 0 h. And the transcription level of *ZmCpn60-3* increased gradually 0–72 h in *Pst* DC3000-treated plants, and its expression was increased by 7.70-fold at 72 h compared to 0 h, (Figure 2f); meanwhile, the expression of *ZmCpn60-3* in *Sclerotinia sclerotiorumi*-treated plants after 24 h were 5.28-fold higher compared to 0 h (Figure 2g). Moreover, the transcription level of *ZmCpn60-3* increased gradually 0–48 h in *Fumonisin* B1-treated plants, and its expression was increased by 1.91-fold at 48 h compared to 0 h (Figure 2h). The above results indicated that *ZmCpn60-3* may be involved in plant disease resistance.

Meanwhile, the expression of *ZmCpn60-3* in different maize tissues was detected by qPCR, and it was found that *ZmCpn60-3* was widely expressed in the stem, followed by the tassel, bracts, and leaves, while the expressions level in the pollen and anther were lower (Figure 2i), indicating that *ZmCpn60-3* is involved in plant growth and development and has certain tissue specificity in its expression.

### 2.3. Overexpression of ZmCpn60-3 in Arabidopsis Increases Resistance to Pst DC3000

To clarify the function of *ZmCpn60-3*, transgenic Arabidopsis overexpressing *ZmCpn60-3* was obtained (Appendix A). Two independent T3 homozygous lines (L4 and L9) were selected to analyze the functional characteristics of *ZmCpn60-3*. The results showed that there was no difference in plant growth between transgenic plants and wild-type plants, indicating that the insertion and overexpression of *ZmCpn60-3* had no effect on the normal growth of Arabidopsis (Figure 3a).

After spraying with *Pst* DC3000, the leaves of different groups showed slight chlorosis and shrinkage at 3 d, and yellow necrosis spots were obvious on plant leaves at 7 d; however, the leaves of transgenic *Arabidopsis* had fewer necrotic spots and a smaller expansion area compared to wild control (Figure 3b). The disease indices of transgenic lines L4 and L9 at 7 d infected with *Pst* DC3000 were 36.47 and 37.93, respectively, which were significantly lower than that of WT (44.34) and empty plasmid-overexpressing *Arabidopsis* (43.48) (Figure 3c). In addition, after inoculation with *Pst* DC3000, the bacterial count in *ZmCpn60-3*-overexpressing lines at different time points were all lower than that in WT and empty plasmid-overexpressing plants. Moreover, the bacterial count in different groups all reached their maximum at 4 d; however, the counts in transgenic lines L4 and L9 were 17.00 × 10^4^ cfu/g and 19.00 × 10^4^ cfu/g, respectively, which were significantly lower than WT (28.33 × 10^4^ cfu/g) and empty plasmid-overexpressing plants (26.33 × 10^4^ cfu/g) (Figure 3d). These results suggested that the expression of *ZmCpn60-3* in *Arabidopsis* inhibited the expansion of *Pst* DC3000 and improved plant resistance.

Histochemical analysis with trypan blue at 3 days post-inoculation (dpi) with *Pst* DC3000 revealed pronounced accumulation of dark blue necrotic lesions at infection sites in transgenic lines compared to wild-type and empty vector-transformed controls (Figure 3g), indicating that the overexpression of *ZmCpn60-3* in *Arabidopsis* could induce a hypersensitive response to resist the invasion of *Pst* DC3000.

The DAB histochemical staining method employs a hydrogen-peroxide-specific oxidation reaction mediated by heme-containing proteins (e.g., peroxidases), generating dark brown deposits that precisely trace the spatial distribution patterns of reactive oxygen species (ROS) within plant cells. The results showed the leaves of the *ZmCpn60-3*-overexpressing plants had significant brown pigment deposition near the *Pst* DC3000 infection site compared with the WT and empty plasmid-overexpressing leaves, suggesting more H_2_O_2_ accumulated in transgenic plants. Quantitative analysis revealed that hydrogen peroxide (H_2_O_2_) levels in all experimental groups rapidly increased following inoculation with *Pst* DC3000. Transgenic lines L4 and L9 exhibited the highest accumulation, reaching 17.88 μmol/g and 14.87 μmol/g, respectively, showing significant increases compared to wild-type plants (10.23 μmol/g) and empty vector-overexpressing controls (10.98 μmol/g) (Figure 3e), suggesting the overexpression of *ZmCpn60-3* in *Arabidopsis* could induce the accumulation of H_2_O_2_ against *Pst* DC3000 infection.

Then, the expression of many key genes, including the *PR1*, *PR2*, *PR5*, NPR1, *LOX2*, *COI1*, *PDF1.2*, and *ERF1* in SA, JA, and ET signaling pathways of *Arabidopsis,* were analyzed to clarify the signaling pathway involving in *ZmCpn60-3*. At 48 h after spraying with *Pst* DC3000, *PR1*, *PR2*, and *PR5* were strongly induced in transgenic lines, and the expression of the above genes in L4 were 2.28, 2.51, and 3.49 times higher than those in wild plants, respectively (Figure 3f). NPR1 and *COI1* were slightly induced, while *LOX2*, *PDF1.2*, and *ERF1* were not induced in transgenic Arabidopsis (Figure 3f). Results revealed that *PR1*, *PR2*, and *PR5* were simultaneously highly expressed in *ZmCpn60-3*-overexpressing lines after *Pst* DC3000 infection, indicating that *ZmCpn60-3* might participate in plant defense mechanisms against *Pst* DC3000 by activating the SA-dependent signaling pathway.

The pCUB-ZmCpn60-3-GFP and pCUB-GFP expression vectors were successfully constructed and transiently expressed in maize protoplasts (Figure 4a). Quantitative analysis demonstrated a 25-fold upregulation of *ZmCpn60-3* gene expression in the pCUB-ZmCpn60-3-GFP-transfected protoplasts compared to the control group (Figure 4b). Then, the expressions of *PR1*, *LOX1*, and *ERF1*, which are key genes in the SA, JA, and ET signaling pathways, were analyzed, and it was found that the expressions of *PR1*, *LOX1*, and *ERF1* in maize pCUB-*ZmCpn60-3*-GFP protoplasts were significantly lower than those in maize pCUB-GFP protoplasts without pathogen inoculation. As shown in Figure 4c, transient expression assays revealed distinct defense-related gene responses in protoplast systems. Following treatment with 0.5 μM flg22 (a *Pst* DC3000-derived elicitor), pCUB-ZmCpn60-3-GFP protoplasts exhibited a marked 5.2-fold greater induction of PR1 expression compared to the pCUB-GFP control. In contrast, transcript levels of ERF1 and LOX1 showed no significant differences between the *ZmCpn60-3*-expressing and control protoplasts under identical elicitation conditions. This differential gene activation pattern strongly suggests *ZmCpn60-3*’s specific involvement in salicylic-acid-mediated defense signaling rather than jasmonic-acid- or ethylene-regulated pathways.

### 2.4. ZmCpn60-3-Overexpressing Arabidopsis Exhibited Resistance to Pst DC3000 Through the SA-Mediated Signaling Pathway

Results indicated that *ZmCpn60-3* might participate in plant defense mechanisms against *Pst* DC3000 through the SA signaling pathway. Thus, changes of endogenous SA and JA contents in *ZmCpn60-3*-overexpressing plants before and after *Pst* DC3000 infection were detected by HPLC-MS/MS. The retention times of the peaks of the SA and JA extracts were consistent with the standard SA and JA, appearing at 7.20 ± 0.06 min and 8.07 ± 0.05 min (Appendix A). The SA contents in transgenic lines and wild-type plants rapidly accumulated at 48 h post *Pst* DC3000 infection when compared to the negative controls. The SA contents in the transgenic lines L4 and L9 reached 200.95 and 179.33 ng/g FW, respectively, which were significantly higher than that in wild plants (71.07 ng/g FW). However, there was no significant difference in JA content between the transgenic lines and the wild plants at 48 h post infection with *Pst* DC3000, and the JA contents in the transgenic line L4 and L9 were only 13.87 and 20.58 ng/g FW, respectively (Figure 4d), while the JA content in the wild plant was 22.32 ng/g FW. The results suggest that the overexpression of *ZmCpn60-3* in *Arabidopsis* could induce the accumulation of SA and inhibit the formation of JA, thus activating the SA signaling pathway against *Pst* DC3000 infection.

**Figure 4 plants-14-01993-f004:**
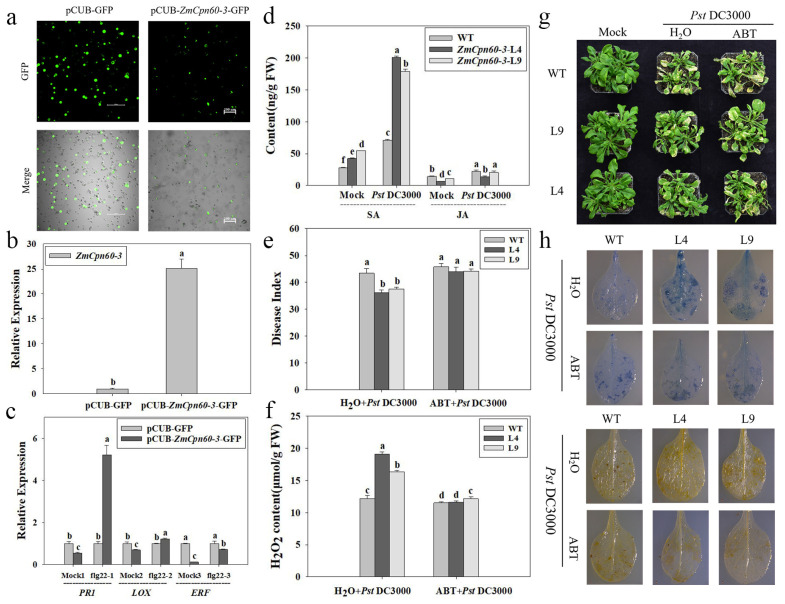
*ZmCpn60-3* enhanced resistance to *Pst* DC3000 through the SA mediated signaling pathway. (**a**) pCUB-ZmCpn60-3-GFP and pCUB-GFP vectors were transiently expressed in maize protoplasts with >50% transformation efficiency. GFP is a green fluorescent protein. Scale bar = 100 μm. (**b**) Expression of *ZmCpn60-3* extracted from maize protoplasts, which expressed pCUB-*ZmCpn60-3*-GFP or pCUB-GFP. (**c**) Expression of the marker genes in pCUB-*ZmCpn60-3*-GFP or pCUB-GFP mazie protoplasts, before and at 2 h post flg22 peptide treatment. *ZmActin* and *ZmGADPH* were used as reference genes for normalization. The maize protoplasts that expressed pCUB-GFP were used as the control and assigned a value of 1. “Mock” and “flag22” represented the expression of the marker genes in pCUB-ZmCpn60-3-GFP or pCUB-GFP maize protoplasts before and after 0.5 μmol/L flg22 peptide treatment, respectively. (**d**) The contents of salicylic acid and jasmonic acid in different groups before and at 48 h post *Pst* DC3000 infection. “Mock” and “*Pst* DC3000” represented the SA and JA contents in different groups before and after *Pst* DC3000 infection. (**e**) Disease indices of the different groups at 7 d post *Pst* DC3000 infection, which were pretreated with ABT for 24 h. (**f**) Detection of H_2_O_2_ content of *Arabidopsis* leaves at 3 d post *Pst* DC3000 infection in different groups, which were pretreated with ABT for 24 h. (**g**) Phenotypes of the different groups at 7 d post *Pst* DC3000 infection, which were pretreated with ABT for 24 h. (**h**) Trypan blue staining and DAB staining of the different groups at 3 d post *Pst* DC3000 infection. Data are presented as the mean ± SD based on three biological replicates, where different lowercase letters indicate significant differences at *p* < 0.05.

To furtherly confirm that *ZmCpn60-3* enhanced *Arabidopsis* resistance to *Pst* DC3000 through the SA signaling pathway, ABT, a SA biosynthesis inhibitor, was used to pretreat the groups with 200 μmol/L for 24 h; then, different groups were inoculated with *Pst* DC3000, and the disease index was calculated at 7 dpi. It was found that transgenic lines L4 and L9, pretreated with ABT, exhibited a large number of necrotic spots (Figure 4g); their disease indices were 44.04 and 44.16, respectively, which were not significant difference from WT (45.80), while the negative transgenic lines L4 and L9 were 36.20 and 37.55, respectively (Figure 4e).

Meanwhile, the necrotic cells in transgenic plant leaves, which were pretreated with ABT, were detected by trypan blue staining. There were fewer necrotic cells around the *Pst* DC3000 infection site in transgenic plant leaves when compared to the negative controls, suggesting that ABT pretreatment inhibited a hypersensitive response in transgenic lines (Figure 4h). Moreover, the brown precipitate in transgenic plants pretreated with ABT was obviously reduced at 3 d infected with *Pst* DC3000 (Figure 4h); the H_2_O_2_ contents in the ABT-pretreated transgenic lines L4 and L9 were 11.62 and 12.16 μmol/g, which were significantly lower than those in the negative transgenic lines L4 (19.07 μmol/g) and L9 (16.37 μmol/g) (Figure 4f). These results indicated that ABT application reduced *ZmCpn60-3*-overexpressing resistance to *Pst* DC3000, conversely indicating that the overexpression of *ZmCpn60-3* in *Arabidopsis* enhanced resistance to *Pst* DC3000 by activating the SA signaling pathway.

### 2.5. Screening of ZmCpn60-3 Interactive Proteins in Maize Using Yeast Two-Hybrid Assay

In this study, a recombinant bait vector was constructed and transformed into the yeast strain Y2HGold to screen the toxicity and self-activation ability of the bait vector, and it was found that the yeast Y2H (pGADT7-T + pGBKT7- ZmCpn60-3), the positive (pGADT7-T + pGBKT7-P53), and the negative (pGADT7-T + pGBKT7-Lam) all formed colonies on the SD/-Trp/-Leu plate (Figure 5a), indicating that bait plasmid pGBKT7-ZmCpn60-3 and empty vector pGADT7 were successfully transformed into yeastY2H. Moreover, only the positive control formed colonies on the SD/-Ade/-His/-Leu/-Trp plate (Figure 5b), which suggested that ZmCpn60-3 had no self-activation.

Then, the potential interacting proteins of ZmCpn60-3 were screened using the maize cDNA library, and the two proteins were obtained; sequencing and blast alignment analysis showed that the two proteins were ZmbHLH118 and ZmBURP7. Further validation results showed that the colonies of the different groups all grew well on the 2D synthetic dropout medium (Figure 5c), but only the positive control (pGADT7-T + pGBKT7-P53), pGBKT7 + AD-ZmbHLH118, and pGBKT7 + AD-ZmBURP7 grew well on the 4D selective medium (Figure 5d). The results indicated that ZmCpn60-3 might interact with ZmbHLH118 and ZmBURP7.

To further validate the interactions between ZmCpn60-3 and ZmBURP7, as well as ZmbHLH118, the recombinant plasmids ZmCpn60-3-cYFP, ZmBURP7-nYFP, and ZmbHLH118-nYFP were co-transformed into *Nicotiana benthamiana* leaves. The result showed that the yellow fluorescent protein (YFP) signals were exclusively localized in the nuclei of *N. benthamiana* leaf cells when ZmCpn60-3-cYFP was co-transformed with both ZmBURP7-nYFP and ZmbHLH118-nYFP (As indicated by the red arrows). In contrast, no fluorescent signal was observed in the control experiment where ZmCpn60-3-cYFP was co-transformed with nYFP alone (Figure 5e). Importantly, when ZmCpn60-3-cYFP was co-transformed with both ZmBURP7-nYFP and ZmbHLH118-nYFP into *N. benthamiana* leaves, yellow fluorescence was detected in the nuclei, indicating that ZmCpn60-3 interacted with both ZmBURP7 and ZmbHLH118 and that this interaction occurred within the nuclei. These findings support the conclusion that ZmCpn60-3 interacted with ZmBURP7 and ZmbHLH118, and the complex localized to the nucleus.

## 3. Discussion

In this study, bioinformatics analysis of *ZmCpn60-3* was firstly conducted, and it was found that *ZmCpn60-3* contained a typical Cpn60_TCP1 chaperone domain, and belonged to the β subunit of Cpn60. Many reports have shown the Cpn60 could keep cellular protein homeostasis, and be involved in the development, growth, and mature stages of life, assisting protein folding, assembly, and membrane transport and guiding protein degradation [14,16]. In recent years, numerous studies have shown Cpn60 plays a more important role in the response of plants or eukaryotes to external abiotic stress [17,30,31]. For example, the *Cpn60α* from *Chlamydomonas rheinis* specifically interacts with type II intron RNA, and thus may participate in the function of the organelle RNA splicing factor [32]. Meanwhile, the temperature-sensitive mutation (ems50-1) is a point mutant of CPN60α2, with amino acid position 399 mutated from G to A. When exposed to low temperatures, less chlorophyll accumulates in the new tissues, resulting in specific disturbances in embryonic photosynthesis [20]. However, there were few reports focusing on the function of *Cpn60* in plant resistance to biotic stress. In our study, we isolated the *ZmCpn60-3* from maize, and found that *ZmCpn60-3* is implicated in the regulation of *Arabidopsis* resistance to *Pst* DC3000; the regulatory effects were achieved by inducing hypersensitivity response, plant oxidative burst activity, and activation of the SA signaling pathway, suggesting that *ZmCpn60-3* participated in plant defense responses. The research results indicated that the *ZmCpn60-3* plays an important role in plant disease resistance processes.

Numerous studies have shown that the Cpn60 is mainly localized in chloroplasts [33]. In this study, subcellular localization prediction conducted via WoLF PSORT (https://wolfpsort.hgc.jp/, accessed on 20 October 2023) suggested that ZmCpn60-3 may localize to the chloroplast, a finding consistent with previous reports. In future experiments, we will verify the predicted results through subcellular localization. Moreover, studies had shown that different subunits of Cpn60 could ensure its electron transport function and enhance the expression of Rubisco activase in plants so as to improve photosynthesis and salt resistance of the plants by affecting the NdhB or NdhH of the NDH complex proteins and protecting the photosystem II [34]. In this study, the overexpression of *ZmCpn60-3* in *Arabidopsis* improved the resistance to *Pst* DC3000 by inducing the SA signaling pathway; however, the photosynthetic characteristics and salt resistance mechanism of *ZmCpn60-3*-overexpressing *Arabidopsis* remain unclear. In subsequent tests, the height and pod bearing of the *ZmCpn60-3*-overexpressing *Arabidopsis* will be measured, and the photosynthetic response curves, chlorophyll fluorescence parameters, and the ability to respond to salt stress of the *ZmCpn60-3*-overexpressing lines will be analyzed, so as to clarify the mechanism of *ZmCpn60-3* affecting the plant growth.

Furthermore, based on the findings, we hypothesized that *ZmCpn60-3* plays an important role in disease resistance through activation of the SA signaling pathway. Salicylic acid (SA), as a plant hormone, plays an important role in enhancing plant immunity by binding to NPR1 oligomers [35] and conditioning an increase in cytosolic reducing power [36]. With further research, It was also found that SA had a negative impact on plant growth [37,38]. SA accumulation in plants may cause dwarfism [39]. SA-mediated growth suppression exhibits tissue and species specificity, as exemplified by rice (*Oryza sativa*) seedlings showing no detectable shoot growth inhibition upon exogenous SA application up to 1 mM, a concentration that typically induces phytotoxicity in dicot species [40], while a concentration as low as 10 μM SA inhibited root growth of *Arabidopsis* seedlings [41,42]. During the experiment, it was found there was no difference between transgenic and wild plants during plant growth (Figure 3a), and since the SA contents in the transgenic *Arabidopsis* line L4 and L9 at 48 h post *Pst* DC3000 infection reached 200.95 and 179.33 ng/g FW, respectively, we speculated the SA accumulation amount did not affect plant growth. In the subsequent tests, we will measure and compare the plant height, fresh weight, and seed-setting rate of transgenic and wild-type plants to clarify the impact of SA accumulation on plant growth.

In this study, the maize cDNA library was screened by yeast two-hybrid assay, and two potential interacting proteins of ZmCpn60-3 were obtained; then, the interactions between the candidate ZmbHLH118, ZmBURP7, and ZmCpn60-3 were further verified by yeast two-hybrid point-to-point assay. Studies have shown that the bHLH and BURP protein families play important roles in response to abiotic stresses in plants [22,38,43]. bHLH, which is one of the largest transcription factor families in plants, can form protein complexes with WD40 and MYB transcription factors, thereby improving plant resistance to environmental stress such as drought, high salinity, and cold [44]. The transcription factor AtbHLH122 in Arabidopsis enhances abiotic stress resilience (drought, NaCl, osmotic) through dual mechanisms: transcriptional repression of CYP707A3 and consequent elevation of abscisic acid (ABA) biosynthesis, thereby modulating ABA homeostasis during stress adaptation [43]. Overexpression of CsbHLH041 in cucumber seedlings significantly increased the activities of antioxidant enzymes SOD, catalase (CAT), and peroxidase (POD) under salt stress and improved plant salt tolerance [38]. In addition, an ICE1-like gene, *IbbHLH79*, was isolated and overexpressed in sweet potato. Transgenic plants overexpressing *IbbHLH79* showed cold tolerance by activating the CBF pathway [45]. Meanwhile, dehydrin 22s (RD22 s) AtRD22, a subfamily of the plant-specific BURP-domain-containing protein family, was reported to be upregulated by water stress, salt stress, and exogenous abscisic acid (ABA) [46], and its induction has been used as a marker for abiotic stress [47,48,49]. Based on the functional reports of bHLH and BURP protein families, to which interacting proteins ZmbHLH118 and ZmBURP7 belong, respectively, we speculated that the *ZmCpn60-3* might be involved in the plant abiotic stress response. In the follow-up experiments, we will further validate the yeast two-hybrid results by the bimolecular fluorescence complementation technique (BiFC), and the interaction mechanism between ZmbHLH118, ZmBURP7, and ZmCpn60-3 and the synergistic effect on plant stress resistance activity will be studied in depth.

## 4. Materials and Methods

### 4.1. Bioinformatics Analysis of ZmCpn60-3

The coding sequence (CDS) and protein sequence of ZmCpn60-3 were downloaded from the Phytozome database (http://phytozome.jgi.doe.gov/pz/portal.html, accessed on 18 March 2023). The conserved motifs in ZmCpn60-3 were detected on the SMART website (http://smart.embl-heidelberg.de/, accessed on 17 March 2023). The obtained sequences were then clustered using the neighbor-joining (NJ) method using MEGA v7.0 software, and the constructed phylogenetic tree was optimized using the ITOL website (https://itol.embl.de/, accessed on 15 May 2024) [50,51]. Subcellular localization prediction of ZmCpn60-3 was conducted via WoLF PSORT (https://wolfpsort.hgc.jp//, accessed on 20 October 2023).

### 4.2. Vector Construction and Arabidopsis Transformation

Primers from both ends of the CDS sequence of *ZmCpn60-3* were designed to obtain its full length, with Blast testing on usability on the NCBI website. The fragment was inserted into the vector pCAMBIA1301 (digested with BamHI and PstI) to obtain the recombinant vector p35S: *ZmCpn60-3*, with the primer sequence as follows: *ZmCpn60-3*-BamHI: GGGGATCCATGGCCAAAATGGTGTTGCTCTGC, *ZmCpn60-3*-PstI: GCCTGCAGACCGCTGCCGCCGAGCA. Subsequently, the correctness of the recombinant plasmid was confirmed through enzyme digestion verification and sequencing methods. T3 homozygous transgenic *Arabidopsis* were produced through *Agrobacterium*-mediated genetic transformation [50].

### 4.3. Pathogen Cultivation

*Pst* DC3000 was cultured in beef protein liquid medium at 28 °C for 24 h, and the bacterial suspension was adjusted with 10 mM MgCl_2_·6H_2_O (OD_600_ = 0.4).

### 4.4. Plant Materials and Treatments

Maize (B73) plants were grown at 28 °C, 16/8 h light/dark cycle, and 60–70% relative humidity. Arabidopsis thaliana was grown at 23 °C, 16/8 h light/dark cycle.

The tissue-specific expression and induction patterns of *ZmCpn60-3* were analyzed. Salicylic acid (SA) (1 mM), jasmonic acid (JA) (50 μM), ethephon (ET) (1 mM), or *Fumonisin* B1 (10 μmol/L), the conidial suspension of *Curvularia lunata* (1 × 10^5^ CFU/mL), *Pantoea stewartii* suspensions (1 × 10^6^ CFU/mL), *Pst* DC3000 suspensions (1 × 10^6^ CFU/mL), and *Sclerotinia sclerotiorum* (an dish with 6 mm diameter) were used to treat maize plants at the trilobal stage, then the leaves, which were at different periods after treatment with pathogens, hormones, or mycotoxin, were sampled to extract and analyzed the induction patterns of *ZmCpn60-3*. Moreover, the different tissues of maize were extracted to clarify the tissue-specific expression of *ZmCpn60-3*. Each group consisted of three replicates (each replicate with three maize plants).

Four-week-old wild and transgenic *Arabidopsis* were sprayed with *Pst* DC3000 suspension (OD_600_ = 0.4), and the negative control was sprayed with 10 mM MgCl_2_·6H_2_O suspension. The disease severity was recorded 7 days after *Pst* DC3000 infection [52] to calculate the disease indices; thirty plants were treated individually and each treatment was divided into three replicates. The leaves of different groups were sampled to extract RNA at 48 h after *Pst* DC3000 infection.

On the 2nd, 4th, and 6th day after infection with *Pst* DC3000, the number of bacteria in the leaf tissues of different groups was determined by the gradient dilution smear method, with 3 replicates per group (3 *Arabidopsis* strains per replicate).

At the same time, to further confirm that *ZmCpn60-3* enhances plant resistance through the SA signaling pathway, 100 μmol/L 1-aminobenzotriazole (ABT) was sprayed on 4-week-old wild-type and transgenic *Arabidopsis*. After 24 h of ABT pretreatment, *Pst* DC3000 suspension (OD_600_ = 0.4) was sprayed, and plants pretreated with sterile distilled water were used as negative controls.

### 4.5. Quantitative PCR

RNA of the plant leaves and maize tissues were extracted according to the method of Mao et al. [53]. Each PCR tube contained 10 μL AceQ qPCR SYBR Green Master Mix, 25 ng cDNA, and 1 μL of each primer (Appendix A), and the thermal cycling conditions were as follows: 94 °C for 30 s, 40 cycles (94 °C for 5 s, 55 °C for 15 s). Data processing was performed as described previously [54].

### 4.6. DAB and Trypan Blue Staining

To detect the extent of H_2_O_2_ accumulation in *ZmCpn60-3*-overexpressing Arabidopsis leaves at 3 dpi treated with *Pst* DC3000, DAB staining was performed on the leaves. At 3 dpi treated with *Pst* DC3000, Arabidopsis leaves were collected and hydrogen peroxide was qualitatively detected by DAB staining as previously described [55]. *Arabidopsis* leaves treated with 10 mM MgCl_2_·6H_2_O were used as the negative controls.

Trypan blue staining was also performed following previously described methods [53]. Leaves infected with *Pst* DC3000 for 3 days were immersed in lactophenol blue solution (Sigma-Aldrich, St. Louis, MO, USA) for overnight reaction. The stained leaves were decolorized in 2.5 g/mL chloral hydrate solution until colorless. Transgenic leaves and wild-type leaves treated with 10 mM MgCl_2_·6H_2_O were used as negative controls for *Pst* DC3000 treatment.

### 4.7. Detection of H_2_O_2_

The wild-type and transgenic Arabidopsis leaves were harvested 3 dpi after *Pst* DC3000 treatment, and the H_2_O_2_ content was determined by titanium sulfate spectrophotometry using a Coming kit (Suzhou Coming Biotechnology Co., Ltd., Suzhou, China). Arabidopsis leaves treated with 10 mM MgCl_2_·6H_2_O were used as negative controls.

### 4.8. Effects of ZmCpn60-3 Overexpression on Resistance Genes in the Maize Protoplast Using the Transient Expression System

After the maize plant (B73) germinated, it was moved to the dark for 14 days and replenished with water. Then, the third leaf was cut to prepare protoplasts [56]. After successfully instantaneously expressing maize protoplasts with pCUB-*ZmCpn60-3*-GFP and pCUB-GFP vectors, respectively, bacterial PAMP flg22 (0.5 μmol/L) incubated maize protoplasts for 2 h. Extracting RNA from corn protoplasts and analyzing the expression of key genes in disease resistance signaling pathways [57] (Appendix A). The maize protoplasts, in which pCUB-GFP vector successfully expressed, were used as the negative controls. The flg22 peptide QRLSTGSRINSAKDDAAGLQIA [58] was synthesized by Tsingke Biotechnology (Nanjing, China) Co., Ltd.

### 4.9. Detection of SA and JA in ZmCpn60-3-Overexpressing Arabidopsis

High-performance liquid chromatography–tandem mass spectrometry (HPLC-MS/MS) was used to monitor the SA and JA contents of wild-type and transgenic Arabidopsis thaliana before and after infection with *Pst* DC3000 for 48 h. The instrument consisted of an Agilent 1290 HPLC system and an MS/MS spectrometer (Applied Biosystems 6500 quadrupole trap) (Agilent, San Francisco, CA, USA) operated in multiple reaction monitoring mode (Nanjing Conversend Testing Technology Co., Ltd., Nanjing, China). The extraction and detection of SA and JA were based on the methods described in the literature [53]. Each group consisted of three replicates (three *Arabidopsis* plants per replicate).

### 4.10. Screening of Interacting Proteins in Maize cDNA Library

The yeast two-hybrid assays were performed as previously described [59]. The CDSs of full-length ZmCpn60-3 were subcloned into the pGBKT7 vector to generate the BD-ZmCpn60-3 construct. ZmbHLH118 and ZmBURP7 were introduced into the pGADT7 vector to generate the AD construct. These BD constructs were co-transformed into the yeast strain Y2HGold with the AD construct. The transformed yeast cells were sprayed with 2D synthetic selection medium (-Trp/-Leu) and 4D medium (-Trp/-Leu/-His/-Ade) and cultured at 28 °C for 3 days. PGADT7-T/PGBKT7-P53 and PGADT7-T/pGBKT7-Lam were used as direct and indirect controls, respectively. The primers used in the vector construction are listed in Appendix A.

### 4.11. Statistical Analysis

TheZmCpn60-3, ZmBURP7, and ZmbHLH118 were ligated into the BIFC vectors pUC-SPYNE and pUC-SPYCE to obtain the recombinant plasmids ZmCpn60-3-cYFP, ZmBURP7-nYFP, and ZmbHLH118-nYFP. The following primers were designed for gene cloning and listed in Appendix A. The empty vectors pUC-SPYNE and pUC-SPYCE served as negative controls. These recombinant plasmids were then transformed into the *A. tumefaciens* GV3101 and co-infiltrated into 4-week-old *Nicotiana benthamiana* leaves. Confocal images were taken 48 h after agroinfiltration at Zeiss Laser Confocal (Jena, Germany).

### 4.12. Bimolecular Fluorescent Complimentary

The statistical significance of the results was calculated using one-way ANOVA followed by the least significant difference Duncan test, with a significant difference level of *p* < 0.05.

## 5. Conclusions

In conclusion, this study identified and characterized *ZmCpn60-3*, a maize chaperonin 60 protein that plays a crucial role in plant defense against biotic stresses. Our results demonstrated that *ZmCpn60-3* is highly expressed in specific maize tissues and can be induced by various hormones, mycotoxins, and pathogens. Overexpression of *ZmCpn60-3* in Arabidopsis enhanced resistance to *Pst DC3000* by enhancing hypersensitive responses, increasing H_2_O_2_ accumulation, and activating the SA signaling pathway. The inhibition of SA biosynthesis significantly reduced the resistance conferred by *ZmCpn60-3*, confirming its SA-dependent mechanism. Furthermore, yeast two-hybrid assays identified ZmbHLH118 and ZmBURP7 as potential interacting partners, suggesting that ZmCpn60-3 may also participate in abiotic stress responses. Overall, this study not only highlights the critical role of ZmCpn60-3 in modulating plant immunity but also provides valuable insights into its potential interactions with stress-related proteins. These findings lay a foundation for further exploration of *ZmCpn60-3* in improving crop resistance to environmental stresses and offer a promising target for developing disease-resistant plant varieties through genetic engineering or breeding strategies.

## Figures and Tables

**Figure 1 plants-14-01993-f001:**
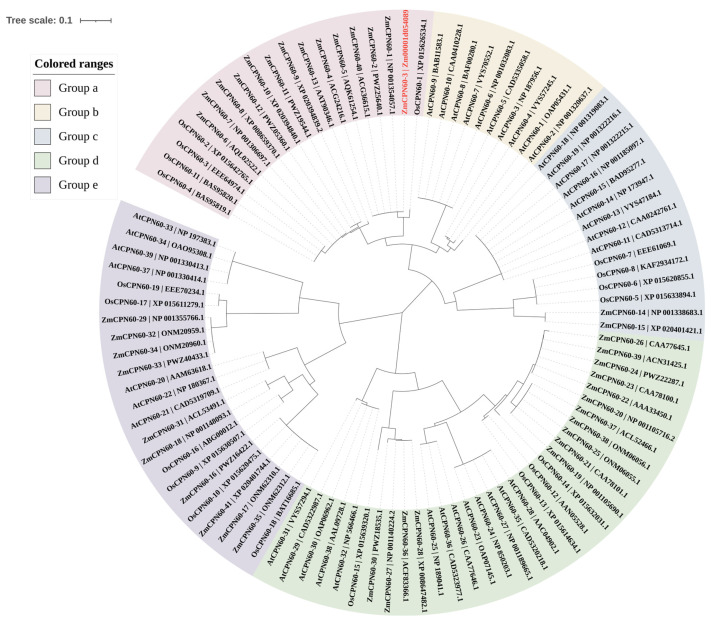
Phylogenetic analysis of the chaperonin protein 60 family from *Arabidopsis*, *Oryza sativa*, and *Zea mays*. The genes encoding the chaperonin protein 60 are divided into 5 categories. *ZmCpn60-3* is marked in red. Bootstrap values of 1000 replicates are shown as percentages at the branch nodes. Bar = 0.1. The protein naming method in the figure was autonomously named with NCBI number.

**Figure 2 plants-14-01993-f002:**
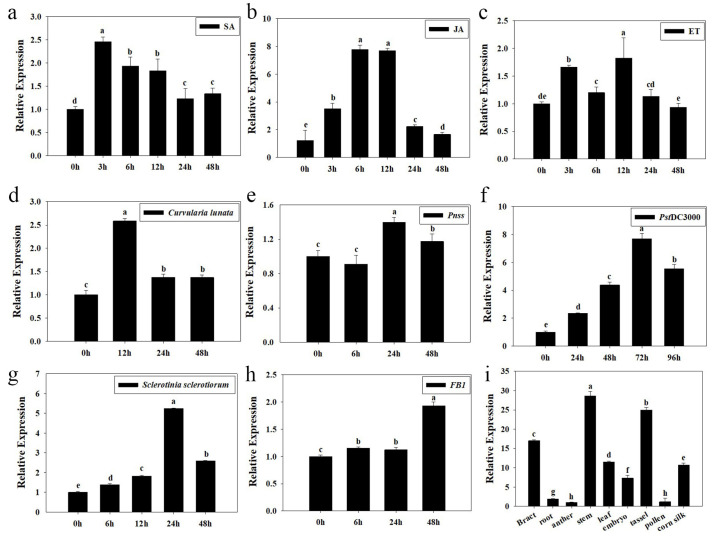
The Induction patterns and tissue expression of *ZmCpn60-3*. (**a**) Salicylic acid (SA) spraying, (**b**) jasmonic acid (JA) spraying, (**c**) ethephon (ET) spraying, (**d**) *Curvularia lunata* (1 × 10^5^ CFU/mL), (**e**) *Pantoea stewartii* (1 × 10^6^ CFU/mL), (**f**) *Pst* DC3000 (1 × 10^6^ CFU/mL), (**g**) *Sclerotinia sclerotiorum* (dish with 6 mm diameter), (**h**) *Fumonisin* B1 (10 μmol/L). For expression normalization, *ZmActin1* and *GAPDH* were employed as dual reference genes, with transcript abundance at the initial time point (0 h) serving as the calibration baseline (normalization factor = 1). (**i**) Tissue expression of *ZmCpn60-3* in maize. Data are presented as the mean ± SD. Three biological replicates were performed, where different lowercase letters indicate significant differences at *p* < 0.05.

**Figure 3 plants-14-01993-f003:**
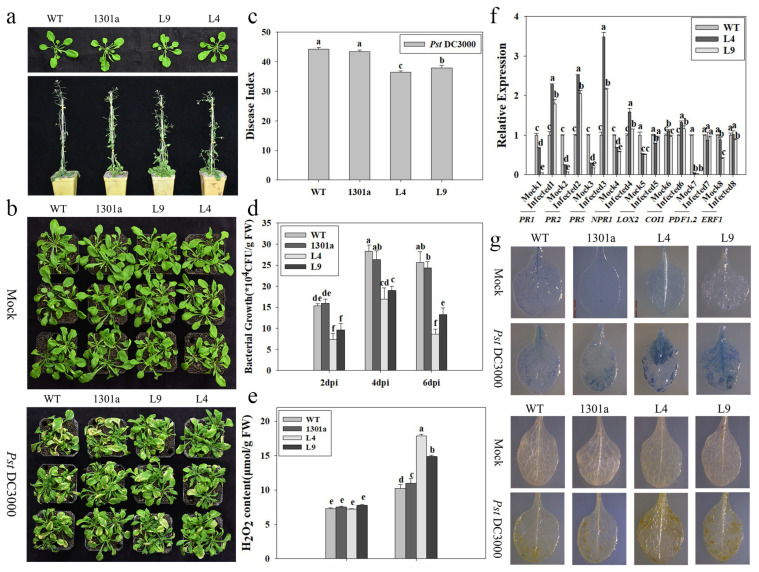
Overexpression of *ZmCpn60-3* in *Arabidopsis* increased resistance to *Pst* DC3000. (**a**) Growth status of Arabidopsis thaliana in different groups at 4 and 7 weeks of age. (**b**) Phenotypes of different groups 7 days after *Pst* DC3000 infection. (**c**) The disease indices of groups at 7 d post *Pst* DC3000 infection. (**d**) The population density of bacteria of different groups infected with *Pst* DC3000. (**e**) H_2_O_2_ content of different groups at 3 d post *Pst* DC3000 infection. (**f**) The expression of target genes in the SA, JA, and ET signaling pathways at 48 h post *Pst* DC3000 infection. *AtActin2* and *AtTUB4* were used as reference genes for normalization. (**g**) Trypan blue staining and DAB staining of different groups at 3 d post *Pst* DC3000 infection. Data are presented as the mean ± SD based on three biological replicates, where different lowercase letters indicate significant differences at *p* < 0.05.

**Figure 5 plants-14-01993-f005:**
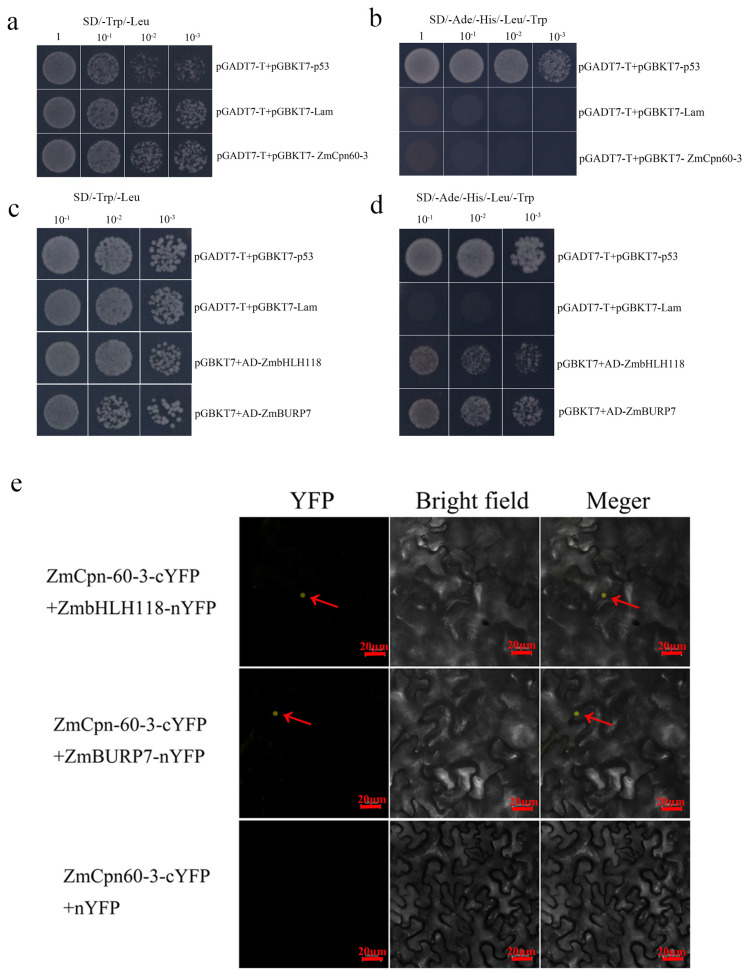
Auto-activation detection and screening the protein interaction with ZmCpn60-3. (**a**) Growth of yeast on SD/-Trp/-Leu plate in self-activation assay. (**b**) Growth of yeast on SD/-Ade/-His/-Leu/-Trp plate in self-activation assay. Yeast two-hybrid screening confirmed the interaction between ZmCpn60-3 and ZmbHLH118 or ZmBURP7. Growth of hybrid yeast on SD/-Trp/-Leu plate (**c**) and SD/-Ade/-His/-Leu/-Trp plate. (**d**) Yeast two-hybrid assays showing the interaction between ZmCpn60-3 and ZmbHLH118 or ZmBURP7. SD/-Trp/-Leu/-His/-Ade medium, (**e**) ZmCpn60-3 interacted with ZmBURP7 and ZmbHLH118 in vivo.

## Data Availability

The authors declare that the data supporting the findings of this study are included in this article. If you need the original data file, you can make a reasonable request to the corresponding author.

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
