# Peer review of "Disease-Resistance Functional Analysis and Screening of Interacting Proteins of ZmCpn60-3, a Chaperonin 60 Protein from Maize"

_plants, 2025, doi:10.3390/plants14131993_

Round 1
Reviewer 1 Report
Comments and Suggestions for Authors
The only problems I saw were based on editing. I have attached a marked-up copy to show areas where changes my be helpful.

see attached file
Author Response
Point1: We will respond to all suggestions on the Abstract section here.
Response: Thank you for the reviewer's suggestions. We have carefully revised the Abstract according to the issues pointed out (Highlighted in yellow; see line 33, 46-47).
Point2: We will respond to all suggestions on the Introduction section here.
Response: Thank you very much for the reviewer's valuable suggestions. We have carefully revised the Introduction according to the issues pointed out (Highlighted in yellow; see line 60-62, 69,75,77-79,81,83,91-94,96,103-104).
Point3: We will respond to all suggestions on the Results section here.
Response: Thank you for your suggestions. We have carefully modified the results according to the problems pointed out (Highlighted in yellow, see line156-157,161,163,188,194,226,271-272,277,281,283-284, 292).
Point4: We will respond to all suggestions on the Discussion section here.
Response: Thanks for the proposed correction. This has been corrected in the manuscript (Highlighted in yellow, see line360,362-363,368-369,378,381,391,396,403,405,429).
Point5: We will respond to all suggestions on the Materials and Method section here.
Response: Many thanks for your comments. We made corrections and they are highlighted in yellow in the manuscript, see line 446-448,463-465,469,513.
Point6: We will respond to all suggestions on the Conclusions section here.
Response: Thanks for the valuable suggestion. We made corrections and they are highlighted in yellow in the manuscript, see line 559.
Reviewer 2 Report
Comments and Suggestions for Authors
The paper is about a Chaperonin 60 protein from maize that the authors clones and named ZmCpn60-3.
They clone the CDS; sequenced it and made a phylogenetic tree. They say it is chloroplastic but did not even do the prediction. They checked the expression in response to hormones or to pathogens.
The authors overexpressed ZmCpn60-3 in Arabidopsis thaliana and characterize the response to biotic cues. The biotic stress phenotyping is well conducted.
To be honest, I am not quite at ease with such overexpression in other organisms… what does that really tell us about the role of the protein in maize?
If they do transformation in Arabidopsis, why no transformation with a GFP fused variant? Why not transform a sid2 mutant, or a NahG?...
They also performed a yeast two-hybrid assay indicating that ZmCpn60-3 might bound to the ZmbHLH118 and ZmBURP7. The authors write it is “preliminary” why is it preliminary. If it is preliminary, can it be published?
Reading is quite ok, but there are many editing mistakes, “,” that should be “.”, italics missing for species and genes and so on….. the authors should take care of that before sending to referees. The name of the pathogens are not written correctly: Genus name always with a first capital letter.
Line 22. Would it not be better to write « proteins » ? idem line 26.
Line 59: “they are mainly divided into two different types, including type I and type II” not clear
Line 65: dalton is D or Da?
Line 80: “chaperone”
Line 81: “,” should be replaced by “.”
Line 82: “,” should be replaced by “.”
Line 83: if you mean gene expression the name of the gene should be in italics
Line 86 : “conditions. [“ delete “.”
Line 88: “,” should be replaced by “.”
Line 89 replace “as” by “an”
Line 92: why “respectively”
Line 92: again a “,” that should be replaced by “.”
Line 95: again a “,” that should be replaced by “.”!
Line 101 to 104: rephrase
Line 104: “In this study,” not clear if this is the current study or the study quoted just before.
Line 116: “the coding region of ZmCpn60-3 was amplified by RT-PCR”: amplified from what. RNA? Extracted from what?
Line 127: “NCBI website” they do that? Are not there sited dedicated to predict localization??
I think there is a problem with the figure 1 embedded in the pdf
Line 254 delete “Preliminary”
It could have been nice to overexpress your gene of interest in sid2 mutants.
Line 350 : “In this study, the subcellular localization prediction of ZmCpn60-3 suggested it might be localized in chloroplast,” but what prediction have you made??
Comments on the Quality of English Language
the english is OK, but you need to edit the punctuation
Author Response
Point1:The paper is about a Chaperonin 60 protein from maize that the authors clones and named ZmCpn60-3. They clone the CDS; sequenced it and made a phylogenetic tree. They say it is chloroplastic but did not even do the prediction. They checked the expression in response to hormones or to pathogens. The authors overexpressed ZmCpn60-3 in Arabidopsis thaliana and characterize the response to biotic cues. The biotic stress phenotyping is well conducted.
Response: Thank you for your valuable comments and recognition of our work. We have supplemented and improved it in the manuscript. In detail, we used the internationally recognized online prediction tool Wolf PSORT (https://wolfpsort.hgc.jp/) to predict the subcellular localization of ZmCpn60-3 protein. The prediction results show that ZmCpn60-3 protein is localized in chloroplasts. At the same time, the relevant prediction method has been provided in the manuscript (Highlighted in green; , see line376-378,443-444).
Point2: To be honest, I am not quite at ease with such overexpression in other organisms… what does that really tell us about the role of the protein in maize?
Response: Thank you for your valuable advice. We fully understand your concerns about the limitations of heterologous expression systems. This study chose to conduct an overexpression experiment of the maize ZmCpn60-3 gene in Arabidopsis, mainly based on the following considerations: the genetic transformation cycle of maize is as long as 8-10 months/generation, and the Arabidopsis system can be used to clarify whether ZmCpn60-3 is involved in the regulation of plant disease resistance/susceptibility in a relatively short period of time, providing a key basis for subsequent gene editing and overexpression studies of ZmCpn60-3 in maize. In addition, this study found that after pathogen infection, the ZmCpn60-3 Arabidopsis overexpression strain can induce the expression of disease resistance genes related to the salicylic acid signaling pathway in plants. It is worth noting that this result is consistent with the experimental results of ZmCpn60-3 in the maize protoplast transient expression system (Figure 4D). These findings provide an important theoretical basis for the subsequent in-depth analysis of the molecular mechanism of ZmCpn60-3 in regulating maize response to external disease stress.
Point3: If they do transformation in Arabidopsis, why no transformation with a GFP fused variant? Why not transform a sid2 mutant, or a NahG?...
Response: Many thanks for your careful evaluation of our paper. Regarding the GFP fusion vector: This study used the pCAMBIA1301a vector for genetic transformation of Arabidopsis. The vector itself contains the GUS reporter gene system, and the transgenic lines have been effectively identified in the experiment by GUS histochemical staining and qRT-PCR. GUS staining plays a similar reporting function to GFP in indicating transgenic plants and expression patterns. If subsequent studies require precise positioning, we will consider using GFP fusion vectors for supplementary analysis. Regarding the use of sid2 mutants or NahG transgenic materials: The main goal of this study is to clarify whether ZmCpn60-3 is involved in regulating the disease resistance/susceptibility function of plants. The experimental results showed that overexpression of ZmCpn60-3 in Arabidopsis can significantly activate the expression of disease resistance genes related to the salicylic acid signaling pathway. This provides a basis for the subsequent research on ZmCpn60-3 gene editing and overexpression in maize. Therefore, based on the phased research goals, we have not yet used Arabidopsis sid2 and other mutants to deeply analyze the ZmCpn60-3 disease resistance signaling pathway. According to expert opinions, in subsequent studies, we will carry out ZmCpn60-3 gene editing and overexpression corn plant construction, while conducting functional verification of ZmCpn60-3 in Arabidopsis sid2 mutants, in order to systematically clarify its induced disease resistance signal transduction mechanism.
Point4: They also performed a yeast two-hybrid assay indicating that ZmCpn60-3 might bound to the ZmbHLH118 and ZmBURP7. The authors write it is “preliminary” why is it preliminary. If it is preliminary, can it be published?
Response: Thank you for your valuable comments. Yeast two-hybrid is a method to verify the interaction, and there may be false positives. Therefore, we used the expression of preliminary verification in the paper. While the article was being edited and submitted, we also carried out further BIFC verification. The results are shown in Figure 5. The experimental results clearly show that ZmCpn60-3 interacts with ZmbHLH118 and ZmBURP7. The BIFC results section adds (Highlighted in green; see line 334-346, 351-352) that have been placed in the results of the article and (Highlighted in green;see line 541-549) in the materials and methods. The primers used in the experiment are in Appendix 4. All changes have been marked with green background.
Point5: Reading is quite ok, but there are many editing mistakes, “,” that should be “.”, italics missing for species and genes and so on….. the authors should take care of that before sending to referees. The name of the pathogens are not written correctly: Genus name always with a first capital letter.
Response: We are thankful to your positive feedback. The species names and gene names that are not italicized have been corrected as requested, specifically in the article (Highlighted in green; see line 117, 120, 138, 159, 246, 422, 651).
Point6: Line 22. Would it not be better to write « proteins»? idem line 26.
Response: Thank you for your valuable comments. Corrected in the manuscript (Highlighted in green; see line 22, 27).
Point7: Line 59: “they are mainly divided into two different types, including type I and type II” not clear
Response: Thank you for your suggestion, corrected (Highlighted in green; see line 60-61).
Point8: Line 65: dalton is D or Da?
Response: Thank you for your suggestion, corrected (Highlighted in green; see line 66).
Point9: Line 80: “chaperone”
Response: Thank you for your suggestion, corrected (Highlighted in green; see line 81).
Point10: Line 81: “,” should be replaced by “.”
Response: Thank you for your correction suggestion, it has been corrected highlighted in green; see line 82.
Point11: Line 82: “,” should be replaced by “.”
Response: Thank you for your suggestion, corrected (Highlighted in green; see line 83).
Point12: Line 83: if you mean gene expression the name of the gene should be in italics
Response: Thank you for your suggestion, corrected (Highlighted in green; see line 84).
Point13: Line 86 : “conditions. [“ delete “.”
Response: Thank you for your suggestion, corrected (Highlighted in green; see line 87).
Point14: Line 88: “,” should be replaced by “.”
Response: Thank you for your suggestion, corrected (Highlighted in green; see line 89).
Point15: Line 89 replace “as” by “an”
Response: Thank you for your suggestion, corrected (Highlighted in green; see line 90).
Point16: Line 92: why “respectively”
Response: Thank you for your suggestion, corrected (Highlighted in green; see line 93).
Point17: Line 92: again a “,” that should be replaced by “.”
Response: Thank you for your suggestion, corrected (Highlighted in green; see line 93).
Point18: Line 95: again a “,” that should be replaced by “.”!
Response: Thank you for your suggestion, corrected (Highlighted in green; see line 96).
Point19: Line 101 to 104: rephrase
Response: Many thanks for your careful evaluation of our paper. We have made revisions in the manuscript (Highlighted in green; see line 103-106).
Point20: Line 104: “In this study,” not clear if this is the current study or the study quoted just before.
Response: Many thanks for your comments. The phrase "In this study" refers specifically to the research described in our current manuscript.
Point21: Line 116: “the coding region of ZmCpn60-3 was amplified by RT-PCR”: amplified from what. RNA? Extracted from what?
Response: We thank the experts for their valuable comments. The RT-PCR was performed using RNA extracted from maize and cDNA after reverse transcription as template. We have made corrections in the manuscript highlighted in green; see line118-119.
Point22: Line 127: “NCBI website” they do that? Are not there sited dedicated to predict localization??
I think there is a problem with the figure 1 embedded in the pdf
Response: We thank the experts for their valuable comments. We have added the prediction website to the article and added a description in the Materials and Methods (Highlighted in green, see line 443-444). The article also added the subcellular localization prediction by WoLF PSORT (https://wolfpsort.hgc.jp/) which showed that ZmCpn60-3 is localized in chloroplasts (Highlighted in green; see line134-136) Figure 1 has been corrected.
Point23: Line 254 delete “Preliminary”
Response: Thanks for the proposed correction. This has been corrected in the manuscript, highlighted in green; see line 264.
Point24: It could have been nice to overexpress your gene of interest in sid2 mutants.
Response: We thank the experts for their valuable suggestions. According to the experts' suggestions, in the follow-up studies, we will promote the gene editing and overexpression experiments of ZmCpn60-3, and at the same time carry out the functional verification of ZmCpn60-3 in the Arabidopsis sid2 mutant, so as to systematically clarify the disease resistance signal transduction mechanism induced by it.
Point25: Line 350 : “In this study, the subcellular localization prediction of ZmCpn60-3 suggested it might be localized in chloroplast,” but what prediction have you made??
Response: Thanks for the proposed correction. This has been corrected in the manuscript, highlighted in green; see line 376-378.
Round 2
Reviewer 2 Report
Comments and Suggestions for Authors
The authors took into consideration some of my remarks
I appreciate they introduced BiFC data, even though I do not inderstand the results (no colors)
In figure 2 you coud increase the font
the same for fig 3C to 3G
In figure 4C, genes should be italicized
In figure 5, you mean "merged" I do not see any colors.... is that normal?
Author Response
Point1:The The authors took into consideration some of my remarks
I appreciate they introduced BiFC data, even though I do not inderstand the results (no colors)
Response: Thank you for your valuable comments and recognition of our work. The yellow signal in the BiFC experiment is not obvious, which may be due to the strong background light during shooting. For easy observation, we have marked the relevant signals with red arrows in the figure and added green background to highlight the corresponding positions in the text (Highlighted in green; see line 339).
Point2: In figure 2 you coud increase the font
Response: Thanks for the valuable suggestion. We have made revisions in the manuscript (see Figure 2).
Point3: the same for fig 3C to 3G
Response: Thank you for your suggestion,We have made corrections as suggested by the reviewers (see Figure 3).
Point4: In figure 4C, genes should be italicized
Response: Many thanks for your careful evaluation of our paper. We have corrected it (see Figure 4C).
Point5: In figure 5, you mean "merged" I do not see any colors.... is that normal?
Response: Thank you for your suggestion,merged images are superimposed images of bright field and fluorescence channels taken by laser confocal microscopy. The yellow fluorescence signal is weak due to the overly strong background light setting. For easy observation, we have marked the target signal with a red arrow in the figure(Highlighted in green; see line 339).